# WNT Co-Receptor LRP6 Is Critical for Zygotic Genome Activation and Embryonic Developmental Potential by Interacting with Oviductal Paracrine Ligand WNT2

**DOI:** 10.3390/genes14040891

**Published:** 2023-04-10

**Authors:** Fusheng Yao, Jia Hao, Zhaochen Wang, Meiqiang Chu, Jingyu Zhang, Guangyin Xi, Zhenni Zhang, Lei An, Jianhui Tian

**Affiliations:** State Key Laboratory of Animal Biotech Breeding, National Engineering Laboratory for Animal Breeding, Key Laboratory of Animal Genetics, Breeding and Reproduction of the Ministry of Agriculture and Rural Affairs, College of Animal Science and Technology, China Agricultural University, No. 2 Yuanmingyuan West Road, Beijing 100193, China

**Keywords:** WNT signaling, LRP6, WNT2, preimplantation embryos, zygotic genome activation, in vitro fertilization

## Abstract

Mammalian preimplantation development depends on the interaction between embryonic autocrine and maternal paracrine signaling. Despite the robust independence of preimplantation embryos, oviductal factors are thought to be critical to pregnancy success. However, how oviductal factors regulate embryonic development and the underlying mechanism remain unknown. In the present study, focusing on WNT signaling, which has been reported to be essential for developmental reprogramming after fertilization, we analyzed the receptor-ligand repertoire of preimplantation embryonic WNT signaling, and identified that the WNT co-receptor LRP6 is necessary for early cleavage and has a prolonged effect on preimplantation development. LRP6 inhibition significantly impeded zygotic genome activation and disrupted relevant epigenetic reprogramming. Focusing on the potential oviductal WNT ligands, we found WNT2 as the candidate interacting with embryonic LRP6. More importantly, we found that WNT2 supplementation in culture medium significantly promoted zygotic genome activation (ZGA) and improved blastocyst formation and quality following in vitro fertilization (IVF). In addition, WNT2 supplementation significantly improved implantation rate and pregnancy outcomes following embryo transfer. Collectively, our findings not only provide novel insight into how maternal factors regulate preimplantation development through maternal-embryonic communication, but they also propose a promising strategy for improving current IVF systems.

## 1. Introduction

Mammalian preimplantation development is orchestrated through a series of consecutive cellular and molecular events that are finetuned via the synergistic interaction of embryonic autocrine and maternal paracrine signaling. The interaction involves a wide range of growth factor ligands secreted from the embryo or the oviduct, as well as their interacting receptors expressed in preimplantation embryos [1,2].

Embryo-maternal communication through autocrine or paracrine factors initiates from the earliest stages of embryonic development. The robust independence of preimplantation embryo development in vitro suggests that the developmental program is autonomous [2]. However, increasing evidence from our own [3,4] and other groups [5,6] has challenged this conclusion: although the essential role of embryonic autocrine factors in supporting preimplantation development has been well accepted, oviductal paracrine factors are also critical for developmental potential of preimplantation embryos. Despite the importance of the synergistic cooperation of autocrine and paracrine factors in preimplantation development, however, how they cooperate and modulate downstream intracellular pathways during this critical developmental window remains an open question for the developmental biology.

Among the signaling pathways that are essential for cellular survival and growth, canonical WNT signaling plays a critical role in acquiring development potential [7,8]. Activation of canonical WNT signaling involves both Frizzleds (Fzds) and its co-receptor low-density lipoprotein receptor-related protein5/6 (LRP5/6) [9,10]. Autocrine or paracrine WNT ligands can bind to Fzds or LRP5/6, resulting in β-catenin stabilization and accumulation. β-catenin is involved in transcription upregulation by binding to transcriptional factors or to a component of SWI/SNF, a chromatin-remodeling complex that regulates chromatin accessibility [11,12]. When WNT signaling is inactivated due to the absence of extracellular ligands, β-catenin is phosphorylated by glycogen synthase kinase 3 (GSK3) and is ubiquitylated, thus leading to rapid proteasomal degradation. In contrast, when WNT signaling is activated via ligand-induced receptor phosphorylation, β-catenin ubiquitination is prevented, and then, stabilized β-catenin can translocate into the nucleus and act as a coactivator of transcription factors [13]. Although the developmental roles of many ligands and receptors of WNT signaling have been identified in early embryos [14,15], the receptor–ligand repertoire and coupled intracellular pathways remain largely unknown. In particular, the role of oviductal WNT ligands in modulating embryonic WNT signaling has never been determined.

In the present study, we aimed to profile the ligand-receptor repertoire of WNT signaling during preimplantation development and found that embryonic WNT co-receptor LRP6 and its interacting oviductal ligand WNT2 may be the candidates critical for modulating WNT signaling during early development. To test this hypothesis, we functionally detected the developmental role of LRP6 and WNT2, as well as their effect on landmark molecular and cellular events during preimplantation development. Our study will not only present novel insight into understanding embryo–maternal interactions during preimplantation development but will also provide a new strategy for improving current in vitro fertilization (IVF) systems.

## 2. Results

### 2.1. WNT Co-Receptor LRP6 Is Critical to Early Cleavage of Preimplantation Development

To decipher the receptor–ligand repertoire of embryonic WNT signaling, we first analyzed the dynamic expression patterns of WNT signaling receptors in in vivo conceived embryos and their temporally corresponding oviduct/uterus, using our previously published RNA-seq data [16]. Several WNT signaling receptors, such as *Fzd5*, *Fzd6*, etc., showed higher expression levels in oviduct/uterus, indicating that these receptors may function in the oviduct/uterus (Figure 1A). By contrast, *Fzd2*, *Fzd9* and *Lrp6* were primarily expressed in embryos. Of note, *Fzd9* and *Lrp6* were specifically expressed in embryos, and *Lrp6* showed a consistently higher expression throughout preimplantation development. We also validated LRP6 expression on the protein level using immunofluorescent analysis (Figure 1B).

Next, we focused on the stage-specific developmental role of LRP6 in preimplantation embryos by using salinomycin, a potent selective LRP6 inhibitor that can block WNT-induced LRP6 phosphorylation and cause degradation of LRP6 protein [17]. Salinomycin supplementation showed a dose-dependent effect on compromising the preimplantation developmental rate; 20 and 40 μM salinomycin significantly decreased the ratio of embryos that developed to the blastocyst stage and increased the proportion of embryos arrested at the two-cell stage (Figure 1C,D). LRP6 inhibition from the four-cell stage onward did not affect embryo development until blastocyst formation, and LRP6 inhibition from either the eight-cell or morular stage onward has no impact on preimplantation development (Figure 1E–G). These results suggest that LRP6 may be preferentially critical to early cleavage of preimplantation development, thus affecting subsequent blastocyst formation. 

### 2.2. LRP6 Inhibition Impedes Zygotic Genome Activation and Disrupts Relevant Epigenetic Reprogramming

Having confirmed the critical role of LRP6 in early cleavage, we next asked if LRP6 is essential for zygotic genome activation (ZGA), a hallmark event that occurs around the early cleavage stage and is of prime importance for subsequent development. Mouse ZGA includes minor and major waves, which occur before and after second-round DNA replication, respectively [18]. To test if minor or major ZGA impairment was responsible for LRP6 inhibition-induced two-cell arrest, we designed a set of experiments based on the stage-specific salinomycin exposure (Figure 2A). Compared with genetic knockout or knockdown, chemical-induced inhibition can ensure a transient but not consistent or prolonged blockage of LRP6, thus determining its functional window. Using 5-ethynyl uridine (EU) incorporation assay, which specifically labels nascent RNA from de novo transcription [19,20], we found that LRP6 inhibition from the zygote to late two-cell stage significantly impeded both minor and major ZGA (Experiments 1 and 2) (Figure 2B,C). We next showed that LRP6 inhibition-induced ZGA impairment was reversible when embryos were transferred to salinomycin-free medium before 5-EU incorporation (Experiments 3 and 4) (Figure 2D,E). This finding is in line with the developmental consequence of salinomycin exposure at the corresponding stage. LRP6 inhibition that covered minor ZGA resulted in a 50% reduction in blastocyst rate, while the LRP6 inhibition that covered both minor and major ZGA resulted in severe developmental arrest at the two-cell stage, as well as almost complete failure of subsequent development, similar with the consistent exposure throughout the preimplantation stage (Figure 2F,G). Given that WNT signaling plays an important role in modulating various histone epigenetic modifications, such as H3K4me3, H3K9me3, etc. [21], which are also the prerequisites for transcriptional regulation of ZGA [22,23], we next asked if salinomycin-induced two-cell arrest was associated with disrupted histone modifications. Immunofluorescent analyses showed that salinomycin-treated two-cell embryos exhibited aberrantly high levels of H3K4me3 and H3K9me3 (Figure 2H,I), suggesting that LRP6 may play an important role in prompting ZGA via modulating epigenetic reprogramming.

### 2.3. Oviductal Paracrine WNT2 Is Critical for Activating LRP6 and Prompting ZGA

Having confirmed the role of LRP6 in regulating ZGA and supporting preimplantation development, we next attempted to determine the effect of oviductal paracrine factors in activating WNT-LRP6 signaling and prompting embryonic development. To this end, we first compared the activity of WNT signaling between in vivo (IVO) preimplantation embryos and their counterparts generated under standardized in vitro fertilization (IVF) conditions, which were used as the model that lacks oviductal paracrine ligands because currently used commercial culture medium does not include any WNT ligands [24]. By gene set variation analysis (GSVA), we found that IVO embryos exhibited high-level WNT signaling at the early cleavage and blastocyst stages. By contrast, IVF embryos showed lower activity of WNT signaling at these stages (Figure 3A). Gene set enrichment analysis (GSEA) also supports this result. WNT signaling tended to be enriched in IVO embryos (Figure 3B). Then, we attempted to determine the paracrine ligands from the oviduct/uterus that could interact with LRP6 to activate WNT signaling. By integrating UNIPROT-derived secreted proteins and the STRING database, we screened out seven candidate ligands that could interact with LRP6, including WNT3, DKK2, WNT2, etc. (Figure 3C). All seven ligands were specifically expressed in the oviduct/uterus (Figure 3D). Next, we selected WNT2 as the candidate because its expression was more enriched at the early cleavage and blastocyst stages, which temporally coincided with WNT signaling dynamics during preimplantation development. To explore the developmental role of oviductal paracrine WNT, we supplemented WNT2 in culture medium throughout the preimplantation stage, based on mouse embryo assay (MEA) by using in vivo fertilized and in vitro cultured embryos. Although having no obvious impact on embryo preimplantation developmental rate, WNT2 supplementation showed a dose-dependent effect on improving increasing inner cell mass (ICM) cells and ICM:TE (trophectoderm) ratio of blastocysts (Figure 3E–I). Of note, we also found that WNT2 supplementation significantly promoted transcriptional upregulation of both minor and major ZGA, but not in blastocyst activation (Figure 3J–L), a process that is critical for blastocyst–uterine communication and blastocyst implantation [25]. These results indicate that oviductal paracrine WNT2 has an important role in promoting embryonic ZGA, probably via the interaction with its membrane receptor LRP6.

### 2.4. Supplementation of Oviductal Paracrine WNT2 Improves Embryo Quality and Pregnancy Outcomes

Given that oviductal paracrine WNT2 can promote both minor and major ZGA, we next explored the effect of exogenous WNT2 supplementation on the developmental potential of IVF embryos. Partially distinct from the results of MEA, we found that 50 ng/mL WNT2 significantly promoted blastocyst formation (Figure 4A,B), and the beneficial effect was completely abrogated by blocking LRP6 (Figure 4C). Moreover, immunofluorescence analyses showed that WNT2 supplementation has no effect on total cell number, but it increased the ICM cell number and ICM:TE ratio (Figure 4D–G).

Furthermore, we evaluated the subsequent embryo implantation potential using in vitro embryo outgrowth, a model for evaluating embryo implantation potential [26]. Embryos exposed to WNT2 throughout the preimplantation stage showed significantly accelerated blastocyst hatching at 12 h. However, the percentage of blastocysts that attached to and outgrew on the fibronectin, as well as the final outgrowth area, were similar to that in the control group (Figure 4H–L). Finally, we confirmed the beneficial effect of WNT2 supplementation on subsequent pregnancy outcomes following embryo transfer. WNT2 supplementation throughout in vitro culture significantly increased both the implantation and live birth rates. Importantly, WNT2 supplementation did not lead to any detectable fetal or placental defects (Table 1). These results suggest that WNT2 supplementation has a positive role in preimplantation development and subsequent pregnancy outcomes but no detectable adverse effect on fetal growth and health. 

## 3. Discussion

Mammalian preimplantation embryos undergo a series of consecutive events, such as ZGA, cellular division, lineage specification, etc., all of which are essential for subsequent development and pregnancy success. These processes were tightly regulated by multiple signal transduction pathways, and each pathway participates in one or several related biological events [27]. Among these pathways, canonical WNT signaling has been reported to play an important role in influencing the developmental potential of preimplantation embryos [7,15]. However, mechanisms responsible for the modulation of embryonic WNT signaling activity, in particular, how oviductal paracrine ligands regulate embryonic WNT signaling during preimplantation development, remain elusive.

Our results provided the evidence supporting the essential role of LRP6, as the critical receptor of WNT signaling, in determining preimplantation developmental consequence (Figure 1). Of note, using the model of stage-specific LRP6 inhibition, we identified that activation of LRP6 is necessary for minor and major ZGA (Figure 2A–E). LRP6 inhibition covering both minor and major ZGA resulted in a large proportion of two-cell arrest, while LRP6 inhibition that only covered minor ZGA led to a considerable decrease in blastocyst formation (Figure 2F,G). This finding was in line with previously reported developmental consequences following transient inhibition of minor ZGA by DRB, a reversible inhibitor of Pol II-mediated transcription [18]. These facts indicate that LRP6-mediated WNT signaling activation plays an important role in ZGA and is the prerequisite for subsequent development.

Changes in histone modifications are hallmark reprogramming events during the process of ZGA, and disruption in histone reprogramming could directly impair ZGA [18,28,29]. In our study, LRP6 inhibition resulted in disruption of both H3K4me3 and H3K9me3 reprogramming at the two-cell stage (Figure 2H–I). As an active histone mark, H3K4me3 generally appears as a sharp status at the promoter regions to facilitate transcription factor incorporation by recruiting CHD1, a histone remodeler, to “loosen” the chromatin structure [30,31]. In mature oocytes, the genome is highly enriched with non-canonical flat H3K4me3 domains, which can inhibit deposition of H3K27ac and combing of transcriptional factors [22]. The removal of non-canonical flat H3K4me3 was thought to be necessary for the re-establishment of canonical sharp H3K4me3 peaks at the promoter regions, which are critical for triggering ZGA [15,32]. Our results suggest that LRP6 inhibition impeded the removal of oocyte-deposited non-canonical H3K4me3 (Figure 2H). Similarly, another histone modification, H3K9me3, also showed higher levels due to LRP6 inhibition (Figure 2I). As a repressive histone mark, H3K9me3 is tightly involved in the formation of heterochromatin and maintenance of gene silencing in mature oocytes [33]. Thus, post-fertilization removal of H3K9me3 is crucial for initiating ZGA and epigenetic reprogramming that confers developmental potential [34,35]. These results, taken together, imply that LRP6-mediated WNT signaling may participate in the removal of oocyte-deposited histone marks that are necessary for ZGA initiation soon after fertilization.

Highlighting the important role of paracrine factors from the maternal oviduct, we also propose a strategy for improving current IVF systems by supplementing exogenous WNT2 in culture medium. We found that exogenous supplementation of WNT2 not only promoted blastocyst formation (Figure 4A,B), but also improved embryonic lineage commitment, revealed by an increased ratio of ICM:TE (Figure 4D–G). More importantly, we also showed that WNT2 supplementation could increase the pregnancy success rates and final pregnancy outcomes after embryo transfer (Table 1). Our concept is in line with the results of previous studies: epigenomic and gene expression patterns of IVF embryos can be partially corrected by supplementing culture medium with oviductal fluid [36,37]. However, considering the potential risk of disease transmission, the strategy of oviductal fluid addition may be practical to in vitro embryo production in domestic and laboratory animals. By contrast, the chemically defined culture medium using the functional growth factors or cytokines that are present in oviductal fluid should be a more reasonable strategy, especially in the context of clinical use of human-assisted reproductive technologies.

Collectively, based on our results, we propose the model to illustrate the important role of LRP6 and its interacting oviductal ligand WNT2 in supporting preimplantation development. Our data proved that WNT co-receptor LRP6 is necessary for early cleavage and has a prolonged effect on preimplantation development. Activation of LRP6 through oviductal or embryonic WNTs promotes ZGA by affecting H3K4me3 and H3K9me3 modifications (Figure 5, left panel). In contrast, the absence of paracrine WNT ligands in in vitro culture medium may contribute to impaired embryo developmental potential, and aberrant H3K4me3 and H3K9me3 modifications and impeded ZGA may participate in this impairment (Figure 5, right panel).

Overall, focusing on the maternal paracrine factors and their embryonic interacting receptors, our study not only provides novel insight into the critical role and mechanism of the oviductal WNT ligand in finetuning preimplantation developmental reprogramming, but it also supports the concept highlighting the usage of oviductal cytokines or growth factors in increasing pregnancy success rates of IVF embryos.

## 4. Materials and Methods

### 4.1. Animals

Eight-week-old ICR mice were kept in a temperature-controlled room with 12 h alternating light/dark. All mice experiments were approved by the Institutional Animal Care and Use Committee of China Agricultural University. The ethical clearance approval number is AW11107102-1-2.

### 4.2. Oocyte Source, In Vitro Fertilization (IVF) and Embryo Culture

Approximately 600 female mice were super-ovulated by peritoneal injection with 5 IU pregnant mare serum gonadotropin (PMSG, Ningbo, China) and 5 IU human chorionic gonadotrophin (HCG, Ningbo, China) after 48 h. For IVC, the super-ovulated female mice were cocaged individually with male mice after HCG injection. The next morning, mice with vaginal plug were identified as successfully mated and were sacrificed to collect embryo–cumulus complexes. To isolate fertilized zygotes, the embryo–cumulus complexes were treated with 300 mg/mL of hyaluronidase to disperse the cumulus cells, after being washed in M2 medium, the zygotes with pronuclear were transferred to the potassium simplex optimized medium containing amino acids (KSOM + AA; Millipore, Darmstadt, Germany) for culture.

For IVF, the sperm were released from cauda epididymis and capacitated for 1 h in modified Krebs–Ringer bicarbonate medium (TYH). Female mice were sacrificed, and cumulus–oocyte complexes (COCs) were pulled out from oviduct and transferred to modified human tubal fluid (mHTF) for 30 min. Then, sperm were added to mHTF medium for fertilization. After 4 h of incubation, embryos were washed in mHTF and then transferred into KSOM for culture. Early 2-cell, late 2-cell and blastocysts were collected at 22, 29 and 96 h post insemination (hpi).

Unless otherwise stated, embryos were cultured in KSOM medium containing WNT2 (H00007472-P01, Abnova, China) or salinomycin (S8129, SELLECK, Houston, TX, USA) through preimplantation period.

### 4.3. Analysis of Transcription Activity

Embryos were incubated with KSOM containing 500 μM 5-ethynyl uridine (EU; RiboBio, Guangzhou, China) for 1 h at 37 °C and fixed in 4% paraformaldehyde (PFA) for 30 min. Then, embryos were permeabilized in 0.5% Triton X-100 (Sigma-Aldrich, St. Louis, MO, USA) for 20 min and incubated with Cell-Light^TM^ Apollo 488 Imaging Kit (Thermo Fisher Scientific, Waltham, MA, USA) for 30 min. After being washed in 0.5% Triton X-100, then treated for another 10 min. in 0.1% PBS-PVA. The embryos were incubated with 4′,6-diamidino-2-phenylindole (DAPI) for 15 min and mounted on glass slides. Images were captured by using fluorescence microscope (BX51TRF; Olympus, Tokyo, Japan) at a wavelength of 488 nm. The fluorescence intensity was quantified by using Image J software (https://imagej.net/ij/index.html (accessed on 1 January 2023), Rawak Software Inc., Stuttgart, Germany).

### 4.4. Embryo Transfer

On the day of IVF, approximately 60 pseudo-pregnant female mice were mated with vasectomized male mice. The next morning, mice with vaginal plug were recipients and consider as day 0.5. For embryo transfer, six well-developed blastocysts were transferred to each uterine horn of recipient. At embryonic day (E)19.5 (14 days after embryo transfer), recipients were sacrificed to evaluate the development ability of each embryo.

### 4.5. Blastocyst Outgrowth

First, 24-well plates were coated with 500 μL PBS containing 10 μg/mL fibronectin and incubated at 37 °C for 4 h. Then, wells were washed two times with PBS, and Dulbecco’s Modified Eagle’s medium (DMEM; 11965092, Waltham, MA, USA) containing 5% fetal bovine serum (FBS; Gibco) was added to wells for further equilibration at 37 °C for 4 h. At 96 hpi, well-developed blastocysts were transferred to wells and incubated for 48 h. The hatching rate, adhesion rate and outgrowth rate were calculated at 12, 24, 36 and 48 h following transfer to the outgrowth plate. Images of the outgrowing embryos were captured with an inverted microscope (IX71, Olympus, Tokyo, Japan) at 48 h. The area covered by the embryo was recorded as the outgrowth area.

### 4.6. Immunofluorescence Staining

Embryos were washed three times in 0.1%PBS-PVA, and the zona pellucida was removed by acidic Tyrode’s solution (T1788, Sigma, St. Louis, MO, USA), and then, embryos were transferred to 4% PFA for 1 h. After being washed with 0.5% Triton X-100, embryos were blocked in 1% BSA for 1 h and incubated with primary antibodies overnight at 4 °C. Followed wash three times in 0.5% Triton X-100, the embryos were incubated with secondary antibodies for 1 h. Finally, the embryos were incubated with DAPI for 15 min to and fluorescence signals were observed under BX51 microscope (Olympus, Tokyo, Japan).

The following primary antibodies used in this research are listed as follows: anti-LRP6 (1:500 dilution, T58345S, Abmart, Shanghai, China), anti-NANOG (1:500 dilution, ab80892, Abcam, Cambridge, UK), anti-CDX2 (1:500 dilution, MU392A-UC, BioGenex Laboratories, San Francisco, CA, USA), anti-H3K4me3 (1:500 dilution, ab8580, Abcam) and anti-H3K9me3 (1:1000 dilution, ab8898, Abcam).

### 4.7. GSVA and GSEA Analysis

WNT signaling correlated gene set was downloaded from KEGG databases (KEGG: Kyoto Encyclopedia of Genes and Genomes). GSVA analysis was performed by R package of GSVA, and GraphPad was used to visualize the result (https://www.graphpad.com/features, accessed on 1 January 2023). GSEA was performed by using javaGSEA software (https://www.gsea-msigdb.org/gsea/index.jsp, accessed on 1 January 2023), and R package ggplot2 was used to visualize these results.

### 4.8. Statistical Analysis

Two-tailed *t* test was used to analyze significant differences between the two groups. For multiple comparisons, one-way ANOVA was used to compare differences among groups by using IBM SPSS 18; significance was set as * for *p* < 0.05, ** for *p* < 0.01, *** for *p* < 0.001.

## Figures and Tables

**Figure 1 genes-14-00891-f001:**
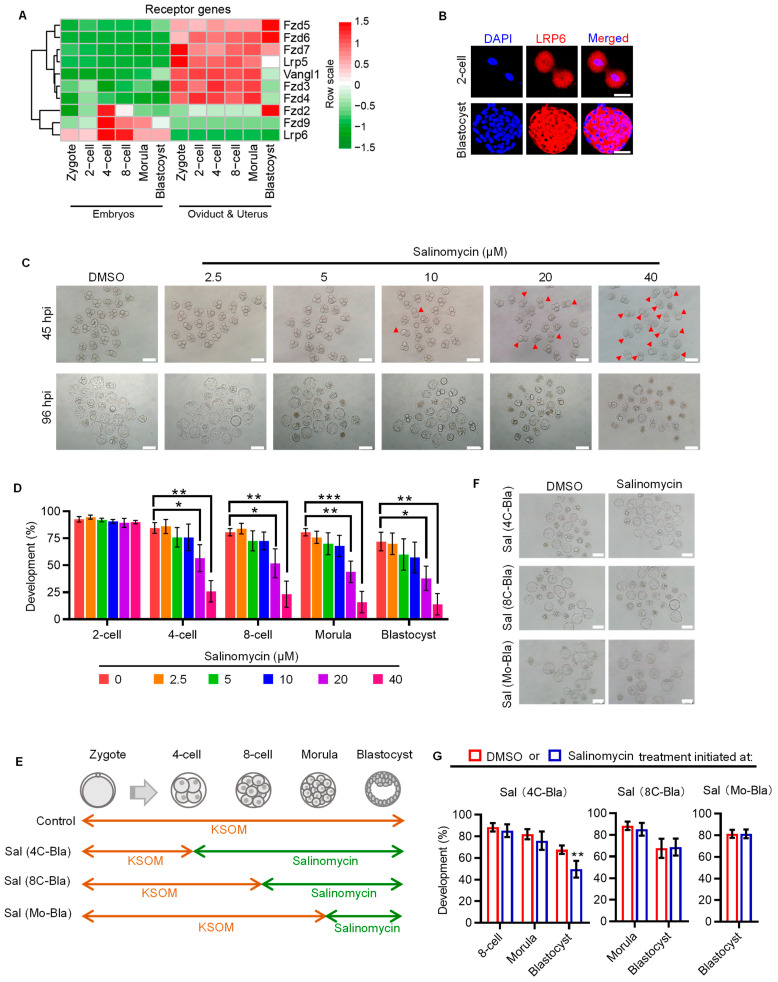
LRP6 plays an important role in early cleavage stages. (**A**) Heatmap showing the dynamic expression patterns of receptors of WNT signaling in IVO embryos and corresponding oviductal and uterus tissues. (**B**) Representative images of LRP6 staining in 2-cell and blastocyst. Scale bar = 50 μm. (**C**) Representative images of 4-cell and blastocyst following culture from zygotes in DMSO or salinomycin. Red arrows indicated arrested 2-cell. Salinomycin is a potential LRP6 inhibitor. hpi, hours post insemination. Scale bar = 200 μm. (**D**) Developmental progression of IVF preimplantation treated with or without salinomycin. * *p* < 0.05, ** *p* < 0.01, *** *p* < 0.001. (**E**) Experimental design of embryo culture treated with or without salinomycin initiated at 4-cell, 8-cell and morula. Sal, salinomycin; 4C, 4-cell; 8C, 8-cell; Mo, morula; Bla, blastocyst. Related to (**F**,**G**). (**F**) Representative images of blastocysts following culture of 4-cell, 8-cell and morula stage embryos in 40 µM salinomycin. Scale bar = 200 μm. (**G**) Percentage of embryos to reach the various preimplantation embryo stages following culture of 4-cell, 8-cell and morula stage embryos in 40 µM salinomycin. ** *p* < 0.01.

**Figure 2 genes-14-00891-f002:**
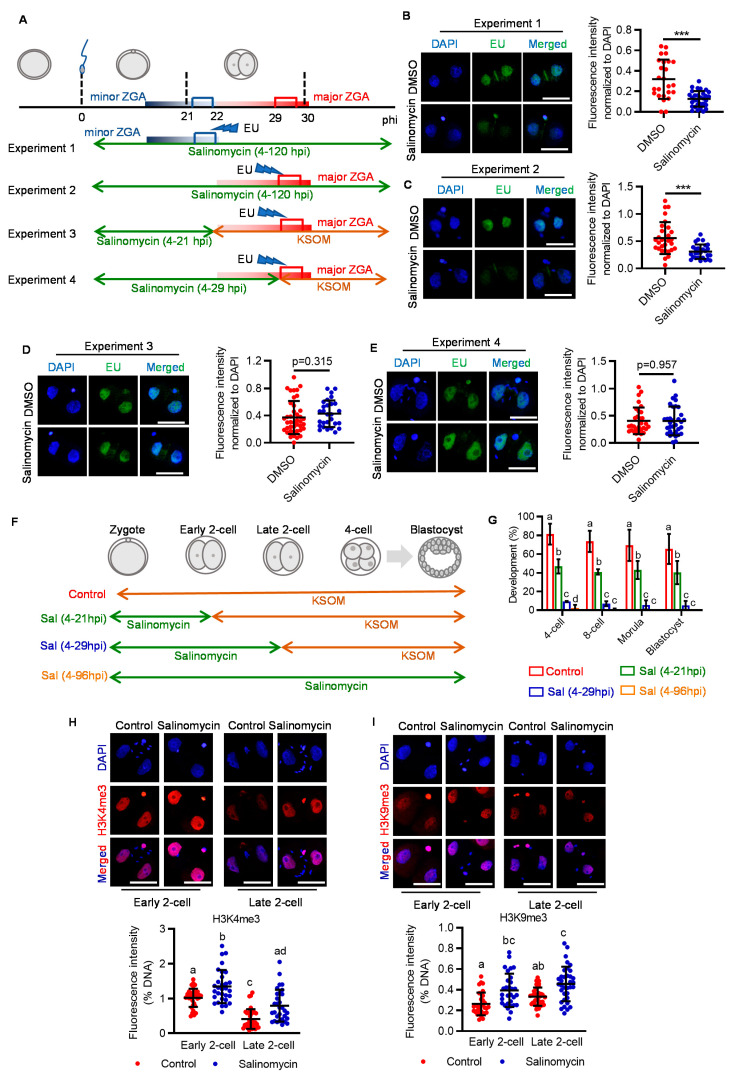
Inhibition of LRP6 impairs ZGA. (**A**) Experimental design of EU incorporation in salinomycin-treated embryos at various periods. (**B**–**E**) Representative images of EU staining at 2-cell embryos (**upper panel**) and quantification of the EU signal intensity (**lower panel**). Each plot indicates relative staining intensity in each embryo. Scale bar = 50 μm. Error bar = means ± SD. *** *p* < 0.001. (**F**) Experimental design of salinomycin addition at different time points. (**G**) Development rate of embryos to reach the various preimplantation embryo stages. Scale bar = 50 μm. Error bar = means ± SD. (**H**,**I**) Representative images of H3K4me3 (H, **upper panel**) and H3K9me3 ((**I**) **upper panel**) staining in early or late 2-cell treated with or without salinomycin. Scatter plot showing the quantification of relative H3K4me3 (H, **lower panel**) and H3K9me3 ((**I**) **lower panel**) signal intensity. Scale bar = 50 μm. Error bar = means ± SD. Different letters (a, b, c, d) indicate a significant difference (*p* < 0.05).

**Figure 3 genes-14-00891-f003:**
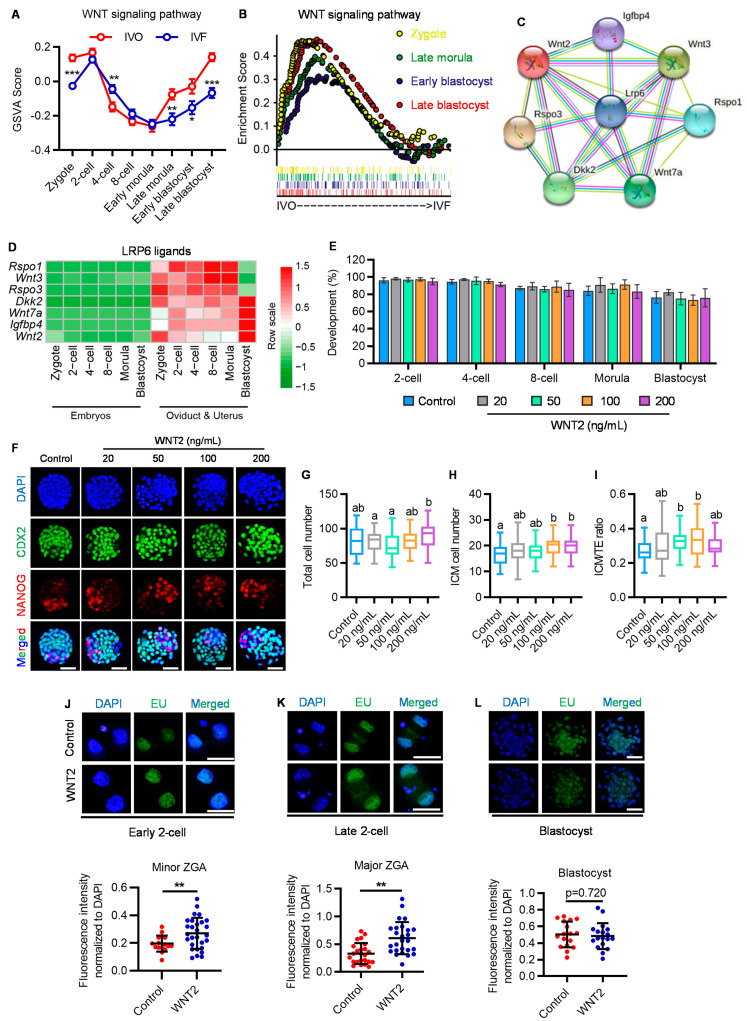
Exogenous WNT2 promotes embryo development via interacting with LRP6. (**A**) GSVA plot showing the dynamic activity of WNT signaling in IVO and IVF preimplantation embryos. * *p* < 0.05, ** *p* < 0.01, *** *p* < 0.001. (**B**) GSEA plot showing the enrichment of WNT signaling pathway in IVO and IVF preimplantation embryos. (**C**) The interaction network of LRP6 with its ligands based on STRING database. (**D**) Heatmap showing the dynamic expression patterns of LRP6 ligands in IVO preimplantation embryos and corresponding oviductal and uterus tissues. (**E**) Development rate of embryos at different stages treated with or without WNT2. (**F**) Representative images of CDX2 (green), NANOG (red) and DAPI (blue) staining in control and WNT2-exposed IVF blastocysts. Scale bar = 50 μm. (**G**–**I**) Quantification of total cell number (**G**), ICM cell number (**H**) and ICM:TE ratio (**I**) of blastocysts is indicated; each plot represents one blastocyst. Related to Figure 3F. Different letters (a, b) indicate a significant difference (*p* < 0.05). (**J**–**L**) Representative images of EU staining in WNT2 treated or without treated early 2-cell ((**J**) **upper panel**), late 2-cell. ((**K**) **upper panel**) and blastocyst ((**L**) **upper panel**). Scatter plot showing the relative signal intensity of EU staining. Scale bar = 50 μm. Error bar = means ± SD. ** *p* < 0.01.

**Figure 4 genes-14-00891-f004:**
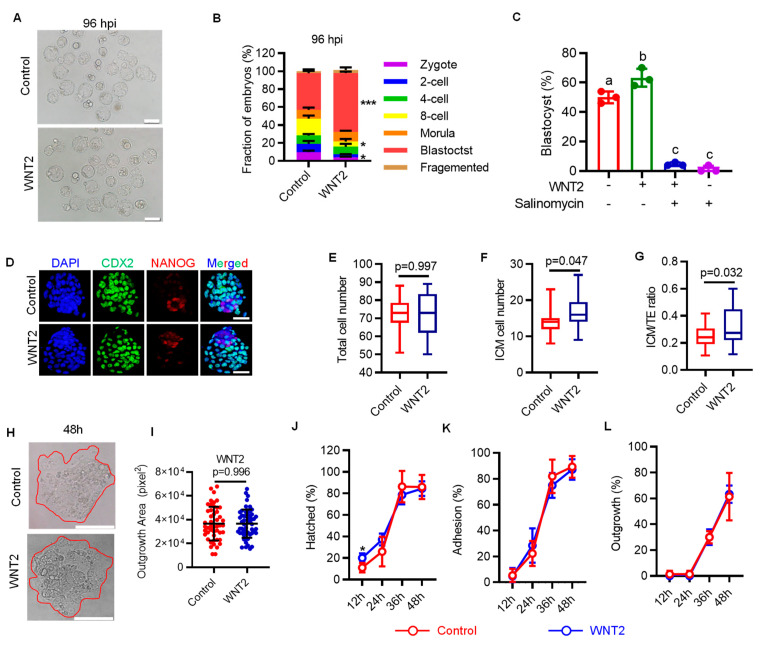
Exogenous WNT2 increases the implantation potentials of embryo. (**A**) Representative images of IVF blastocysts treated with or without WNT2 from zygote stage. Scale bar = 200 μm. (**B**) Fraction of embryos at blastocyst stage after treated with or without WNT2. Error bar = means + SD. * *p* < 0.05, *** *p* < 0.001. (**C**) Effect of combing WNT2 and salinomycin on embryo development. Data are shown as means ± SD. Different letters (a, b, c) indicate a significant difference (*p* < 0.05). (**D**) Representative images of CDX2 (green), NANOG (red) and DAPI (blue) staining in control and WNT2-exposed IVF blastocysts. Scale bar = 50 μm. (**E**–**G**) Quantification of total cell number (**E**), ICM cell number (**F**) and ICM:TE ratio (**G**) of blastocysts, each plot represents one blastocyst. Error bar = means ± SD. Related to Figure 4C. (**H**,**I**) Representative images of IVF blastocysts after 48 h in vitro culture (**H**) and quantification of outgrowth area (**I**). Red circle represents outgrowth. Scale bar = 200 μm. Error bar = means ± SD. (**J**–**L**) The ratio of IVF blastocysts hatched (**J**), adhesion (**K**) and outgrowth (**L**) after 12, 24, 36 and 48 h in vitro culture. Error bar = means ± SD. * *p* < 0.05.

**Figure 5 genes-14-00891-f005:**
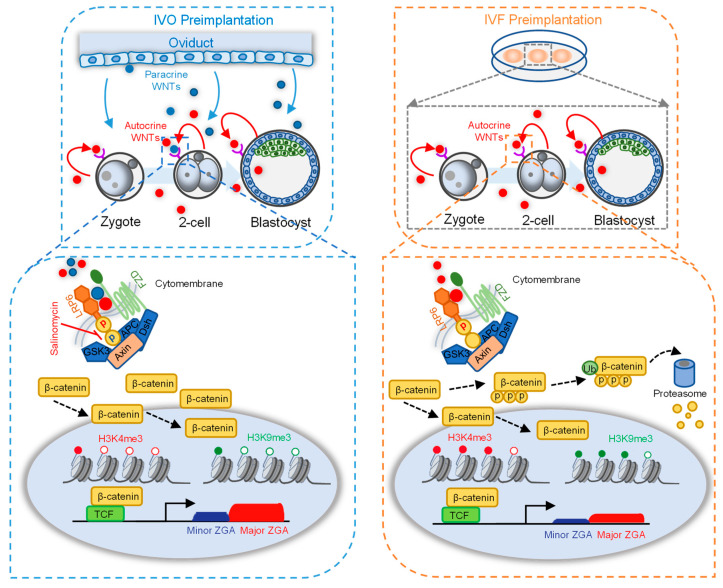
A model illustrating the important roles of LRP6 and paracrine ligands in preimplantation development. The absence of paracrine ligands from the oviduct/uterus in in vitro culture medium impaired embryo developmental potential with aberrant high levels of H3K4me3 and H3K9me3 modification at the ZGA period. Activation of LRP6 through oviductal and embryonic WNTs promotes in vivo ZGA by affecting H3K4me3 and H3K9me3 reprogramming (**left panel**). The inactivation of LRP6 in the absence of paracrine WNTs from the oviduct/uterus in vitro culture medium impaired ZGA accompanied with aberrant high levels of H3K4me3 and H3K9me3 modification (**right panel**).

**Table 1 genes-14-00891-t001:** Pregnancy outcomes of IVF blastocysts treated with WNT2 at E19.5. The mice were sacrificed at E19.5, and the fetuses were collected to measure all indicators.

	Control	WNT2
Recipients (*n*)	31	18
Implantations (%)	66.13 ± 23.37	82.87 ± 15.33 *
Fetuses (% of transferred)	30.11 ± 13.50	39.81 ± 17.25 *
Fetuses (% of implanted)	50.56 ± 23.60	48.42 ± 19.74
Fetal weight (g)	1.54 ± 0.30	1.56 ± 0.27
Occipito-frontal diameter (mm)	6.79 ± 1.26	6.88 ± 1.38
Crown rump length (mm)	24.44 ± 2.22	24.80 ± 2.21
Tail length (mm)	12.61 ± 1.79	12.90 ± 1.25
Placental weight (g)	0.16 ± 0.03	0.15 ± 0.04
Placental diameter (mm)	8.89 ± 1.17	8.63 ± 1.10

Data are shown as means ± SD. * *p* < 0.05.

## Data Availability

The data presented in this study are available on request from the corresponding author.

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
