# Peer review of "WNT Co-Receptor LRP6 Is Critical for Zygotic Genome Activation and Embryonic Developmental Potential by Interacting with Oviductal Paracrine Ligand WNT2"

_genes, 2023, doi:10.3390/genes14040891_

Round 1

Reviewer 1 Report

Genes-2170423

General comments

The intended journal may choose to publish this manuscript. It follows a logical experimental sequence demonstrating how LRP6 and WNT function. The work consists of a series of experiments that required much labor, as evidenced by the results, particularly in the figure section. It‘s an extensive and densely packed outcome.

The author‘s writing appears to be sloppily done, nevertheless. There are numerous typos and errors in the writing. Please carefully review the manuscript portion for this, then reformat step by step accordingly.

I have highlighted some significant modifications that need to be known in the specific comments. For instance, Figure 5 was a hypothetical model that needed to be elaborated on in the discussion section but better described. Table 1 is also absent, yet it is still listed in the manuscript.

I advise the author to thoroughly read the paper and reconstruct how to present the findings and conclusions. The work is firmly adequate. However, the presentation is the issue. And also, it‘s hard to give specific comments since there is no line number on each page.

Specific comments

·         Title. It‘s too long and confusing; reframe it.

·         The author and affiliation need to be reviewed.

·         Last paragraph of the introduction

·         Please reframe the last sentence into a precise study aim. The written sentence is the study conclusion.

·         Page 3, 3rd paragraph. LRP6 inhibition....... seems incomplete. Please re-check

·         No Table 1 presented in the manuscript

The discussion part.

·         In every result referred to in this part is directed to the specific figure to recall the reader‘s memories. For example, on page 5 , in the paragraph beginning with Our result......please specify which result, figure number or table.

·         I didn‘t find any explanation or mention the Figure 5 in manuscript body

Material and method

·         Animals

o   An ethical clearance approval number needs to be mentioned.

·         IVF, IVC, and embryo collection

o   It should be changed into oocyte source, embryo production (IVC or IVF), then embryo culture.

·         Analysis of transcription activity

o   No precise transcription which analyzed. The method explained the fixation and staining protocol. Fluorescence intensity must be mentioned in the wavelength of the ray used.

Reviewer 2 Report

The main question addressed by this research is how during preimplantation development oviductal factors regulate embryonic WNT signalling. I consider that the topic of this study is original and relevant in the field. The authors, in addition to many discoveries supports in their research the concept highlighting the usage of oviductal cytokines or growth factors in increasing pregnancy success rates of IVF embryos. But I believe that at the end of the introduction, the authors, instead of immediately describing their achievements, should specify in detail the goals of their research. My only doubt is whether authors should disclose the number of mice sacrificed for research? Concluding my discourse, for the reader unfamiliar with the presented field of research, it may be difficult to understand the meaning of some abbreviations that have not been fully explained by the authors.

Reviewer 3 Report

The background of the abstract was too lengthy.

In introduction section need rephrases of some sentences as mentioned in attached file.

Hypothesis is missing in the manuscripts.

Grammatical mistake are found in material method section and results and discussion section

Round 2

Reviewer 1 Report

Revised version was better than the original one, and improvement was clearly made